# Nanoindentation and TEM to Study the Cavity Fate after Post-Irradiation Annealing of He Implanted EUROFER97 and EU-ODS EUROFER

**DOI:** 10.3390/mi9120633

**Published:** 2018-11-29

**Authors:** Marcelo Roldán, Pilar Fernández, Joaquín Rams, Fernando José Sánchez, Adrián Gómez-Herrero

**Affiliations:** 1Division of Fusion Technologies, National Fusion Laboratory, CIEMAT, Avda. Complutense 40, 28040 Madrid, Spain; pilar.fernandez@ciemat.es (P.F.); fernandojose.sanchez@ciemat.es (F.J.S.); 2Department of Applied Mathematics, Materials Science and Engineering and Electronic Technology, School of Experimental Sciences and Technology, Rey Juan Carlos University, C/Tulipán s/n, 28933 Móstoles, Spain; joaquin.rams@urjc.es; 3Centro Nacional de Microscopía Electrónica, Universidad Complutense de Madrid, 28040 Madrid, Spain; adriangh@pdi.ucm.es

**Keywords:** nanoindentation, reduced activation ferritic martensitic (RAFM) steels, helium irradiation, irradiation hardening, nuclear fusion structural materials

## Abstract

The effect of post-helium irradiation annealing on bubbles and nanoindentation hardness of two reduced activation ferritic martensitic steels for nuclear fusion applications (EUROFER97 and EU-ODS EUROFER) has been studied. Helium-irradiated EUROFER97 and EU-ODS EUROFER were annealed at 450 °C for 100 h in an argon atmosphere. The samples were tested by nanoindentation and studied by transmission electron microscopy extracting some focused ion beam lamellae containing the whole implanted zone (≈50 µm). A substantial increment in nanoindentation hardness was measured in the area with higher helium content, which was larger in the case of EUROFER97 than in EU-ODS EUROFER. In terms of microstructure defects, while EU-ODS EUROFER showed larger helium bubbles, EUROFER97 experienced the formation of a great population density of them, which means that the mechanism that condition the evolution of cavities for these two materials are different and completely dependent on the microstructure.

## 1. Introduction

The study of the effects of post-irradiation annealing on the evolution of the irradiation defects in which He is involved has a special relevance since its mechanisms and the variables on which those depend are not fully understood, becoming more complicated when the materials investigated are microstructurally complex like the ones studied in this paper. The experimental results referring to nucleation, fate, and consequences of helium irradiation on mechanical properties are especially useful, because they provide information to establish a correlation between experimental results and modelling [1]. In addition, these experiments may also be useful to provide experimental results to validate solubility values and diffusion coefficient of He in its different forms (interstitial atomic He, He-Vacancy clusters (HeV) of different sizes, or even more complex clusters of He and vacancies along with lattice atoms of Fe, HeVFe [2], etc.) which are likewise indispensable when performing modelling work.

In order to validate the theories and models on bubble growth after post-irradiation annealing, it is necessary to examine the effects of some experimental parameters on the evolution of bubbles during annealing by means of experiments under comparable conditions in materials similar to those studied in this research. With the methodology carried out, it has been possible to obtain two structural materials implanted with different levels of He concentration that represent the irradiation expected in a fusion reactor after its useful life (400 to 700 appm) [3,4]. Then, the materials have been annealed at 450 °C, the temperature that has been observed to be critical in the degradation of mechanical properties in materials submitted to irradiation [5]. Numerous authors have indicated at that at 450 °C the greatest volumetric fraction is produced by the formation of cavities generated by irradiation and, therefore, causing the highest degradation of mechanical properties [6,7,8]. Therefore, in order to understand how temperature affects the evolution of cavities for a given time, an annealing treatment has been applied at 450 °C for 100 h in EUROFER97 and EU-ODS EUROFER steels. He desorption experiments with EUROFER97 [9,10] showed that in a temperature range above 450 °C, the dissociation and diffusion phenomena begin to be critical, so to be able to understand the nucleation and growth of He bubbles, it is necessary to evaluate the mechanisms that govern the diffusion of the He atoms prevailing when the implanted materials are undergoing an annealing process, and to try to find a model to frame that behavior. There are different diffusion mechanisms [11] and its relevance depends on factors such as the nature of the He defect, the annealing temperature, and defects present in the material. Each type of defect involving He (Interstitial He, Substitutional He, or HeV clusters) will present different dissociation and migration energies and, therefore, different easiness to move through the crystalline lattice of the material.

A direct consequence of modifying the microstructure of a material is the change of mechanical properties. It is well known that in general terms, He irradiation produces local hardening [12,13,14] and a very useful technique to measure this change is nanoindentation, since He irradiation produces a very shallow layer of modified material. This technique has been widely used to measure hardening due to irradiation [8,15], and it also can be used to establish a correlation between microstructural defects and the aforementioned hardening using a model, such as the dispersed barrier hardening one [16,17]. On one hand, cavity density and size are experimental parameters that can be obtained by TEM (Transmission Electron Microscopy). On the other hand, nanoindentation results evidence the effect of voids, cavities, He bubbles, and dislocations produced after He irradiation as they act as barriers for the dislocations generated by the indenter [18]. So, a model combining both would explain the materials hardening mechanisms.

The reduced activation ferritic martensitic (RAFM) steels EUROFER97 and EU-ODS EUROFER have not been studied extensively after implantation and annealing, although some tests have been performed on similar alloys such as F82H [19]. These materials are very important from the point of view of a nuclear fusion reactor, as they are the most promising structural materials as they are able to withstand the extremely harsh conditions which will be produced in the reactor during operation [20,21]. So, this research is a starting point to understand the growth of the complex defects during the annealing treatment at high temperatures of irradiated steels, which may eliminate the pre-nucleation structure and transforms it into a bubble nucleus (or embryo). The subsequent growth of this embryo or of this already formed bubble will be carried out by means of the mechanisms mentioned above if the necessary conditions of time, temperature, and concentration are met: migration and coalescence [22] or Ostwald ripening [23,24,25,26].

## 2. Experimental Procedure

### 2.1. Materials

The materials investigated in this research were the reduced activation ferritic/martensitic steels EUROFER97 and EU-ODS EUROFER. Both alloys have identical chemical composition (wt. %): 0.11C, 8.7Cr, 1W, 0.10Ta, 0.19V, 0.44Mn, 0.004S, balance Fe. However, EU-ODS EUROFER contains 0.3% of Y_2_O_3_ particles. On one hand, EU-ODS EUROFER has a ferritic matrix with a large range of grain sizes, showing an average value of 0.98 ± 0.48 μm, although it is possible to find grains as large as 4 μm and others smaller than 0.5 μm. An EBSD study was published elsewhere regarding this matter [27]. Yttria particles with 20 nm size in average were added to the matrix, but their distribution was not completely homogeneous, finding both, small clusters of them and large areas with no particles. On the other hand, EUROFER97 presents a fully martensitic matrix, whose primary austenite grain size is between 6.7 and 11 μm and the average size of its martensite laths is between 0.3 and 0.7 μm. In addition, EUROFER97 has equiaxed morphology, in contrast to EUODS EUROFER with ferritic tangle grains. The steels have been studied in the normalized (980 °C/27 min air cooled) plus tempered (760 °C/90 min air cooled) condition for EUROFER97 (Heat E83698) and normalized (1150 °C/60 min air cooled) plus tempered (750 °C/120 min air cooled) for EU-ODS-EUROFER (Heat HXXX1115), denominated in this paper as the as-received states. A more detailed chemical composition and microstructural characteristics for both materials are given elsewhere [28,29].

### 2.2. Irradiation and Thermal Treatment

Ion irradiation with He-ions was performed on a 5 MV voltage terminal in a Tandetron accelerator manufactured by High Voltage Engineering Europa (Amersfoort, The Netherlands) at room temperature. A stair-like profile configuration was used for the implantation energy, starting with 15 MeV and ending with 2 MeV, decreasing the beam energy in 1 MeV steps. The ion fluence used for all the energies was 1.67 × 10^15^ He·cm^−2^. The resulting He concentration profile was simulated by means of MARLOWE Code [30,31] as shown previously [15,27]. The mean damage produced on the sample was estimated to be around 10^−2^ to 10^−3^ dpa, with an average He injection rate of 0.25–0.3 appm He/s.

During implantation, the temperature of the samples was constantly monitored with a thermographic camera and one thermocouple attached to the sample holder. The maximum temperature measured during implantation was 70 °C.

After irradiation, the specimens were annealed at 450 °C for 100 h in a controlled argon atmosphere.

### 2.3. Nanoindentation Tests

The equipment used for nanoindentation was a Nano indenter XP manufactured by MTS Systems Corporation (Eden Prairie, MN, USA) equipped with a Berkovich diamond indenter tip. The indentation module used was the one called Quasi-static (QS). The specimens for nanoindentation tests were mechanically polished, finishing with an acidic aluminum oxide suspension in order to achieve a deformation-free surface.

The indentations on the implanted samples of both steels were performed on the transversal section to the irradiation beam, in order to evaluate the change of the hardness values along of implantation depth to get a hardness profile as a function of the He content after the heat treatment.

He load used was 5 mN, which is the load previously used by Roldan et al. [15,27] to measure the hardness variation on He implanted specimens. However, the methodology to obtain results was improved in terms of time-saving, using in this work only one indentation row tilted 10° with respect to the implantation surface. The distance between consecutive indentations was in all tests above 3 times the imprint size. The Oliver & Pharr method was used to determine the mechanical properties from the indentation curves [32,33].

At the beginning of the study, an analysis of the state of the indenter tip is highly recommended. Figure 1 shows an image obtained by optical profilometer in which it is possible to evaluate the curvature radius of the very tip of the indenter, which is critical for the correct evaluation of the indentation data, especially at shallow depths [34,35]. In this case, the quality of the tip was good enough, just showing some dirt, which is easily removed in fused silica during the alignment of the equipment before making the indentations in the specimens.

### 2.4. Transmission Electron Microscopy

Microstructural investigations on implanted samples of both alloys have been carried out using lamellae extracted by Focus Ion Beam (FIB) in a Zeiss Auriga Compact with a field emission scanning electron microscope (FESEM) of 30 keV. A minimum Ga ion energy of 5 keV was used in the last step of lamella thinning in order to remove as much as possible the Ga damage on the surface [36,37]. Due to the different He concentration along the irradiated volume, it was very important to extract a lamella long enough to allow studying the effect of the annealing with different He concentration, since it is a critical parameter when He bubble fate is studied. Transmission electron microscopy (TEM) investigations were performed with a JEM 2100 HT at 200 keV and JEM 300 F at 300 keV (manufactured by JEOL, Akishima, Japan). In order to calculate a volumetric distribution density of bubbles, it is necessary to determine the thickness of the lamellae, so convergent beam electron diffraction (CBED) was used [38,39,40].

## 3. Results

The studies of EUROFER97 and EU-ODS EUROFER after He implantation previously performed showed a maximum increase of hardness of 41% and 21%, respectively, and corresponded to the zone with the highest He concentration [27]. In the following paragraphs the effect of a post-irradiation annealing of 450 °C during 100 h with regard to nanoindentation hardness and irradiation defects will be described in detail.

### 3.1. Nanoindentation

#### 3.1.1. EUROFER97

The hardness results obtained for EUROFER97 implanted with He from 2 to 15 MeV and consequently annealed at 450 °C for 100 h is represented in Figure 2a in black squares. The dotted red line represents the hardness values for the as implanted condition and the blue dashed line corresponds to the as received state. It is possible to observe that in the annealed state there was a remarkable increase in the hardness values from those of the as received and annealed states, reaching up to 9 GPa, which turns out to be an increase of 157%. There was also a gradual reduction of hardness when the indenting in areas with lower irradiation, reaching the as-received hardness values (~3.5 GPa) at about 25–30 µm from the implantation surface.

#### 3.1.2. EU-ODS EUROFER

In a similar way to EUROFER97, EU-ODS EUROFER was implanted with He from 2 to 15 MeV and annealed at 450 °C for 100 h. The results from the indentations applied with a load of 5 mN are shown in Figure 2b. In this case, the hardness increments experienced are lower than those observed for EUROFER97, since the as received hardness value of EU-ODS EUROFER was 4.6 GPa, and the maximum hardness value measured on the annealed specimen was 8.49 GPa, which results in an increase of 84.5%. However, beyond 30 µm from the implanted surface, the hardness values that diminish from the surface, reached the same hardness values of the as received state. Again, the effect of the annealing is evident as the slight increase observed after irradiation is lower than that observed after the annealing heat treatment.

### 3.2. TEM Characterization

TEM investigations were performed on the implanted surface to up to 30 microns in depth in order to evaluate He nucleation and growth due to the heat treatment, and to eventually correlate those observations with the nanoindentation results obtained.

The observation area was divided in 3 zones, each one with a different He content as observed in Figure 3. Zone A, the one with the highest He content, included the peaks from 3 to 5 MeV (716–657 appm He), zone B included the peaks from 6 to 8 MeV (619 to 548 appm He) and the last zone, C, contained the peaks from 9 to 11 (518 to 445 appm He). It was not possible to study the 2 MeV energy peak, since the first 3 to 5 µm resulted damaged in the fabrication process of the lamella.

#### 3.2.1. Cavity Characterization on EUROFER97

The microstructural study by TEM was performed analyzing a lamella following the implantation direction and, in consequence, along the gradual He content reduction. Figure 4 shows a TEM overview of the lamella extracted from EUROFER97 implanted from 2 to 15 MeV and annealed at 450 °C for 100 h.

The cavities were identified using through-focus series method [41] which in addition allows a first approximation of their size. In the in-focus image, most cavities were not clearly detectable, which means that the cavity size was less than 5 nm, approximately as seen in Figure 5 [42,43]. In the so-called zone A, corresponding to the zone with the peaks of 3 to 5 MeV with a He content of 716 to 675 appm He, the steel matrix of EUROFER97 was completely full of small cavities, which were randomly distributed.

The vast majority of the cavities exhibited a diameter between 1 and 2 nm, and, only in some isolated cases was the diameter slightly larger, but never reaching 3 nm. Taking into account a lamella thickness of around 60 nm, as determined by CBED [38,39], the population density was 9.65 × 10^23^ m^−3^.

The study of cavity distribution did not reveal any preferential nucleation at grain/subgrain boundaries, nor at precipitate–matrix interfaces.

As mentioned at the beginning of this section, if the analysis moves forward following the implantation direction, the implanted He amount decreases gradually up to reach the so-called zone B with a He concentration between 619 to 518 appm. Figure 6 shows cavities found in this zone whose sizes are quite similar to the ones detected at the previous zone, 1 to 2 nm. The cavity distribution was still random and no preferential nucleation was detected. In this zone, the population density was 2.25 × 10^23^ m^−3^, so it was slightly lower than the one measured in zone A, although the cavity average size was quite similar.

In the mentioned figure, the presence of two precipitates type M_23_C_6_ with some cavities attached to their matrix-precipitate (indicated with red arrows in the same figure) can be also observed, and in their surroundings. Due to the lack of zones with no cavities around the precipitates (known as depletion zone), it is not possible to confirm completely that those microstructural features were acting as defect sinks before the annealing. As it has been experimentally observed by TEM [44,45,46], when a microstructural feature acts as a sink, it attracts a HeV cluster flux towards it, creating an area with no defect clusters around the sink. This zone would have certain length that would depend on the vacancy (or HeV cluster) absorption rate which in turn is proportional to the precipitate diffusion and the difference between the defects of the matrix and of the sink surface [47,48].

Finally, to complete the whole TEM characterization of the EUROFER97 lamella, zone C, with He content between 518 and 445 appm He, was analyzed. In Figure 7, it is possible to see in the lower right corner, a black area that was too thick for the electrons to get through. It corresponds with the lamella edge that was welded onto the TEM grid. Unlike A and B areas described previously, groups of very small cavities randomly distributed across the matrix were observed. Some of those clusters have been indicated with arrows and red ovals. The cavity diameters were not larger than 2 nm and the population density was much lower than the one calculated for the other areas, 1.63 × 10^23^ m^−3^. In contrast to zones A and B, some free from cavities areas were detected within this zone. It is difficult to determine exactly where these areas were located regarding the direction of implantation, this fact may suggest that in spite of performing an annealing treatment at 450 °C for 100 h, either the time or temperature were not high enough to induce cavities to diffuse in such a way that they would spread all across the irradiation surface, enhancing nucleation or even the growth of already formed ones, in order to obtain a more homogeneous distribution.

#### 3.2.2. Cavity Characterization on EU-ODS EUROFER

Figure 8 shows an overview of the EU-ODS EUROFER lamella extracted from the implanted and annealed bulk specimen. As can be observed, the lamella was fabricated with some separators highlighted with blue arrows in the figure, with the aim of helping with the identification in TEM of the different areas with different He content, as it is required to correlate microstructure with penetration depth and with He content. The red arrow indicates the ion implantation direction and the zones with different ion implantation dose are marked as A, B, and C.

An exhaustive TEM study was performed in the zone A and in it, some bands of cavities are visible (Figure 9), which may match with a former Bragg’s peak. This area is the closest one to the implantation surface and its width is around 7–8 μm. In addition, in this zone, many cavities randomly distributed across the matrix with a very heterogeneous diameter were detected.

EU-ODS EUROFER present Y_2_O_3_ particles dispersed in the matrix, and due their size and contrast on focus condition in TEM they could be confused with the largest cavities. However, performing the through-focus series, the aforementioned particles’ contrast does not change and cannot be considered in the statistics for cavities.

The size of the cavities was measured, although those with diameters lower than 2 nm were not considered due to the difficulty to measure them accurately (the error may be higher than 15%) and because the volumetric fraction is low in comparison with that of larger cavities [49]. Figure 10 shows the coexistence of cavities of different diameters randomly distributed within the ferritic matrix. In addition, there were not clear indications of cavities at grain boundaries or yttria particles.

The size distribution histogram is plotted in Figure 11. The average size was 4.7 ± 1.2 nm, with cavities even larger than 10 nm, in contrast to EUROFER97 zone A where the average size was 2 nm.

As in the case for EUROFER97, assuming that all the cavities are sphere-like, population density was determined considering only the cavities larger or equal to 2 nm. The calculated value was 1.84 × 10^22^ m^−3^ that corresponds to a volumetric fraction of 0.17%.

The next area of interest, zone B in Figure 8 and with a He content between 619 and 548 appm, is characterized by having a more homogeneous cavity size distribution than the one measured in A, with a Gaussian-like size distribution (Figure 12). Most of the detected cavities presented a diameter between 3 and 4 nm (>35%) and no one with a diameter larger than 7 nm (in contrast to zone A where cavities larger than 10 nm were observed).

Although the cavity distribution seems to be random, within zone B, cavities attached to the Y_2_O_3_–matrix interface and grain boundaries were observed (Figure 13). Those observations may be attributable to the random nucleation process itself that may take place before the annealing and the low mobility during the annealing (high migration energy [50]) of large vacancy-helium clusters. Therefore, it is not possible to assert if the addition of yttria particles under these experimental conditions enhance the defect suppression.

After an exhaustive TEM analysis of different areas belonging to zone B, the distribution density of cavity sizes was calculated following the same procedure than for the zone A, obtaining a value of 3.65 × 10^21^ m^−3^, average size of 3.2 ± 1 nm and a volumetric fraction of 0.12%. In Figure 14 the size distribution of the cavities in zone B is shown.

Finally, in zone C with helium content between 445 to 518 appm, the detection of cavities was more complicated. One of the possible reasons added to the lower implanted helium concentration is that the lamella was slightly thicker in this region than in the previous zones (~70 nm on average). This peculiarity is very common and limits the capability of detecting cavities with diameters smaller than 4 nm.

In zone C, the decrease of cavity density was clearly observed (Figure 15a,b). In general terms, EU-ODS EUROFER presented very similar cavities randomly distributed in the matrix with a size ~2 nm. However, a few larger cavities up to 5 nm were detected as well (highlighted in red in Figure 15a).

Unlike zones A and B, where cavity nucleation was not observed at the grain boundaries (Figure 16), in zone C, some boundaries containing aligned cavities were clearly present. In the right area of Figure 16, two grain boundaries have been highlighted with red ovals, in which a possible alignment of cavities can be observed; they are very similar to those found by Luo et al. [51]. It is difficult to confirm it unequivocally, but as mentioned above, it has been observed by several authors that when grain boundaries act as sinks and trap cavities, an area without cavities appears around these boundaries [48]. In this case, not only there is an area around the grain boundaries free of cavities, but at the right of the red line there appears to be none of relevant size, suggesting that they may be smaller in size, again undetectable by TEM. The value for the distribution density was 9.65 × 10^22^ m^−3^ and the volumetric fraction was 0.002%.

Comparing the results for the three zones it is possible to extract that the lower the helium content, the lower the cavity size, with the most typical cavity size being 4 to 5 nm in zone A, 3 to 4 nm in zone B, and 2 nm in zone C. On the other hand, the maximum sizes observed also have changed, decreasing from 10 nm or larger in zone A to a maximum of 6–7 nm in zone B, whereas in zone C, very few cavities were much larger than 2 nm, however, the maximum size detected was 5 nm.

## 4. Discussion

Regarding nanoindentation results, it has been determined that EUROFER97 and EU-ODS EUROFER steels after irradiation and subsequent annealing at 450 °C experienced a significant increase of hardness. The maximum values are practically identical in both materials, between 8.5 and 9 GPa (Figure 2) measured in the so-called zone A with a high He content. In EU-ODS EUROFER steel, hardness values progressively increase to almost 30 μm depth (~450 appm He), while in EUROFER97 steel, hardness values increase up to a distance of approximately 20 to 30 μm, which is the area of maximum He content (between 400 and 500 appm He). That would correspond in the case of EU-ODS EUROFER steel with the entire lamella depth (zones A, B, and C) and in the case of EUROFER97, the greatest increase occurs mainly in zone A, although also in zone B and to a lesser extent in zone C. The aforementioned increase in hardness in both materials occurs because there are elements that acted as obstacles preventing the sliding of dislocations generated by the indentation. These obstacles are of a very diverse nature: existing dislocations in the material, yttrium oxides (in the case of the EU-ODS EUROFER), precipitates, high quantity of grain edges (small grains), and, of course, the defects produced by irradiation that have evolved due to the annealing treatment, which in this case are HeV clusters, becoming eventually He bubbles (or cavities). When taking into account only the effect of He in the increase of hardness, it is necessary to analyze both the size and the distribution of cavities as possible obstacles for dislocations produced by the indentation volume.

The variation of hardness values of the EU-ODS EUROFER steel is proportional to the size and density of the cavities found in the different areas analyzed. In zone A, the material presents the largest cavity mean size and the widest distribution density, which causes the observed hardness increase. Comparing the results obtained before and after annealing, EU-ODS EUROFER clear change of the irradiations defects observed, since after annealing it experienced a growth of the cavities at the expense of a decrease in distribution density, as is clearly seen in Figure 17a,b.

However, it has been observed that in zone C with a He content between 479 and 445 appm He, EU-ODS EUROFER steels presented some cavities after annealing, not being detected in the window of low implanted He content at room temperature of both materials [27]. Due to the large number of defects (in the form of vacancies and interstitial) present on the EU-ODS EUROFER steel, the thermal diffusion of these defects takes on a special relevance since they move while interacting with the HeV clusters generated by the implantation due to the annealing treatment applied.

In the case of the EUROFER97, a very high increase in zone A (high He content), which has a high density of cavities, was experienced. Figure 17c,d compares the cavities detected in areas with a He content of ~700 appm He in EUROFER97 steel before and after annealing. In this image it is possible to see how the size of the cavities is similar (1 to 2 nm), but the distribution density of these defects from a qualitative point of view seems much higher after the thermal treatment. It should be noted that the annealed steel image was acquired at higher magnifications and, therefore, the cavities appear larger, yet taking a random area in both micrographs results in higher distribution density in the annealed sample.

These hardness values are gradually decreasing, within zones B and C, until they reach values similar to those implanted at room temperature. It should be noted that in the area with the lowest He content of the samples implanted at room temperature without heat treatment no cavities were detected [27], which was the same observation extracted from EUROFER97. However, in the annealed sample there is a substantial increase in population as can be seen in Figure 17d. These results may indicate that due to the effect of temperature there was an increase in nucleation of possible bubble embryos that were detectable by TEM, the mobility of clusters depends completely on their size (the smaller ones present a higher mobility).

Other critical factors are the number of defects in the matrix as well as the microstructural characteristics presented in the material (grain boundaries, precipitates interfaces…). In the case of materials subjected to neutron irradiation [52,53] or another species that generates cascades of displacements, the formation of vacancies is favored by to the formation of Frenkel pairs by irradiation. This leads to a supersaturation of vacancies, since the self-interstitial atoms are very mobile and can form dislocation loops and be trapped by sinks, which will have different effects on the microstructure of the material, and therefore on the mechanical properties. This supersaturation of vacancies increases the diffusivity of He by providing pathways of diffusion [52,54]. In the case of post-irradiation annealing, there was no increase in vacancies due to new cascades, so the parameters that control the evolution of the bubbles are the vacancies already produced by the implantation of He prior to annealing in combination with thermal vacancies, the temperature and thermal treatment duration, He concentration, and, of course, the type of material and its microstructural characteristics (phases, composition, own defects, concentration of vacancies...). With respect to the results mentioned above, it is possible to explain the very different behavior between EU-ODS EUROFER steel and EUROFER97. The first one presents a much higher concentration of vacancies, which, although not measured experimentally, are assumed to be present due to the manufacturing route: powder metallurgy plus the addition of yttria particles. This material has experienced a considerable increase in the size of the cavities in the area with the highest He content (zone A) with respect to the sample of the same concentration but implanted at room temperature [15,27]. In addition, during annealing, the size of the cavities in zones B and C decreases, as well as their distribution density. The chosen temperature activated the diffusion of vacancy clusters of both the material itself and HeV clusters produced by irradiation and favored the reabsorption of these in the already formed bubbles. As the concentration decreases, the amount of nucleated bubbles prior to annealing decreases, and therefore the surface pressure is lower, causing this mechanism to slow down. On the other hand, in EUROFER97, it has been observed that the distribution density increased with respect to implantation at room temperature; however, the average radius remains constant, between 1 and 2 nm. This fact seems to indicate that the decrease in vacancies compared to the EU-ODS EUROFER, implies a decrease in the diffusion process even though the temperature was 450 °C.

Representing cavity size and population density measured in every zone of the implanted depth, it can be observed that EUROFER97 (Figure 18) does not experience cavity growth: in the three zones the diameter is 1 to 2 nm but there is an enormous increase of population density. There was possibly a growth of small bubble embryos (very small HeV clusters).

After the exhaustive analysis, we tried to match the experimental observations to either of the two theoretical model proposed to explain the evolution of He bubbles after post-irradiation annealing: Ostwald ripening [23,24,25,26] and the theory of migration and coalescence (MC) [22].

The Ostwald ripening model, first described in 1896 [26], explains a very common phenomenon occurring in solid or liquid solutions in which homogeneous structures (such as crystals or sol particles) change size over time. This concept has been applied to the cavities formed by HeV clusters, as follows: it is two relatively close cavities of two different sizes formed by an agglomeration of He atoms and vacancies. These structures are energetically stable except for the He atoms that are located on the surface, which are more energetically unstable than those ones inside. As time goes by, these atoms will dissociate from the smaller cavity, causing a decrease in size, having a lower surface energy than the large cavity and will be introduced into the matrix, increasing the number of atoms in solution. When the matrix is oversaturated with He atoms from the cavity that is shrinking, they will be redeposited into the larger cavity around them, causing it to grow. In addition to the dissociation and reabsorption of He atoms, Ostwald ripening also requires the dissociation and reabsorption of vacancies. So, this process which is subject to the dissociation of two species (He atoms and vacancies) will depend on which of the two dissociation energies is larger [55].

On the other hand, the MC theory attributes the growth of the bubbles to the merge of bubbles that, following a diffusion path through the matrix, finally join forming a larger cavity. Ono et al. [56] studied Fe and Fe9Cr samples implanted with He at 10 keV with different fluences and temperatures and applied afterwards an annealing treatment to the samples in steps of the same time from 400 to 1000 °C. In the mentioned paper, it is possible to see the path followed by a mobile cavity when an annealing treatment at 750 °C is applied to a pure Fe sample irradiated with He at 300 °C with a fluence of 6 × 10^19^ ions/cm^2^. This work provides evidence that bubble mobility is Brownian (or random) and thermally activated, as well as experimental evidence that the bubble mobility in Fe9Cr is lower than in pure Fe. This is in agreement with the aforementioned on the importance of alloying elements in terms of cavity mobility. EUROFER97 and EU-ODS EUROFER are not model alloys with a simple microstructure where the cavities can diffuse through the matrix easily, since there are many barriers that prevent the movement. However, in short range order, cavities close to one another may move and coalescence in a more pronounced way when the density of small cavities is as high as that detected in this paper. These observations fit with the results obtained in modelling works regarding the relation between complexity defect and mobility [2,57].

In EU-ODS EUROFER (Figure 18b), the annealing temperature seems to activate the diffusion of vacancies (both thermal and the ones produced by irradiation) which may favor the reabsorption in already formed bubbles. As He content decreases, the number of cavities already formed also diminishes and therefore the internal pressure as well as the number of atoms in solution and vacancies decrease, slowing down the process. These observations may fit Ostwald ripening. Even so, what can be assured is that the vacancies together with the own defects of the EU-ODS EUROFER steel offer preferential sites to form HeV clusters. In addition, the fact that they do not migrate towards grain and/or yttria edges suggests that this is a very local evolution favored by an average temperature. Even so, what can be assured is that the vacancies together with the own defects of the EU-ODS EUROFER steel offer preferential sites to form HeV clusters. There are signs of a growth of cluster embryos, as there is an increase in the density of cavities in the EUROFER97 compared to those implanted at room temperature, but this evolution does not follow either the MC model or the Ostwald maturation model. In the case of the EU-ODS EUROFER, it does appear that the size of the cavities increases with concentration after annealing, with a decrease in density, apparently indicating that Ostwald maturation is favored by a high number of vacancies, according to Trinkaus [52].

Temperature affects the diffusion of defects as observed by modeling. Many authors [2,57] have observed that the mobility increases with temperature, and it is very dependent on the complexity of the defect. Some observations have experimentally demonstrated [48,58,59,60] that the most mobile species are the He interstitial atoms rather than vacancies, even at room temperature. They concluded that alloying with up to an amount in the order of ppm may affect the distribution and size of cavities, thus modifying the distance of the zone around the grain edges without bubbles, which is smaller in purer materials than in commercial ones. Comparing these results with those obtained with EUROFER97 and EU-ODS EUROFER, which present a much more complex microstructure in terms of secondary phases and defects, it has been observed that the alloy elements are not affected as much as the model alloys since both materials present the same atomic concentration of the chemical elements. Therefore, the most notable difference is due to the different microstructures and the manufacturing process. 

By increasing the concentration of He in the material during implantation, there is an increase of the internal pressure. This could explain the different behavior of the bubbles after post-irradiation annealing as there were different He concentration ranges and therefore different surface pressures. In addition, it is known that the surface is a source of vacancies which are available to relax the internal pressure of bubbles close to the surface [55]. Consequently, a slow growth can be attributed to bubbles with high internal pressure, which can increase in size through migration and coalescence. On the other hand, bubbles that grow rapidly can present Ostwald maturation, which seems to present its maximum value when the pressure is relaxed by the vacancies caused by cascades of displacements produced by irradiation or taken from the surface [55]. Stoller et al. [61] observed that as the annealing temperature increased, the size of the bubbles increased and their density decreased instead. They also detected black dots in the matrix that could be mostly small interstitial clusters [62]. Subsequently, Golubov et al. [63] modeled Stoller’s experiment (cited above as [61]) using the theory of bubble migration and coalescence. Without assuming that they remain in mechanical equilibrium during annealing, they concluded that this assumption was quite realistic at high temperatures. Therefore, the bubbles remain over-pressurized during annealing, and this high pressure suppresses surface diffusion and, consequently, growth. On the other hand, by increasing the temperature, the mobility of the vacancy clusters increases, favoring their growth at the same time as their distribution density decreases, as observed experimentally in samples annealed from 700 to 900 °C [61].

On the other hand, having observed zones without cavities, as well as grain boundaries, precipitated interfaces or in the case of the EU-ODS EUROFER, yttria oxides, free of cavities, suggests that reabsorption or growth to be visible in the TEM of new cavities is a short-range, local process, unlike a dislocation network that would be found within the so-called long-range defects. The migration of V-type clusters is carried out by jumps towards the nearest vacancy neighbor, passing through an intermediate metastable state, with its migration energy depending on size as it has been shown above, until reaching V5, when they are considered practically immobile [64]. It is a fact that He implantation damages EU-ODS EUROFER to a lesser extent, since the increase in hardness (which is a direct consequence of irradiation damage) is lower. However, unlike some modelling works [65,66] demonstrating the radiation resistance is only due to yttrium oxide dispersed though the matrix, it is hard to prove since there are other factors such as dislocation density, secondary phases, or inherent vacancies.

## 5. Conclusions

The most important conclusions extracted from this work are the following:The annealing treatment at 450 °C for 100 h has led to an increase in hardness values of 157% for EUROFER97 steel and of 84% for EU-ODS EUROFER with respect to the as-received condition when a load of 5 mN is applied with a Berkovich tip.It was experimentally demonstrated that for faster tests, a row matrix is valid to analyze the surface transverse to the implantation, as long as it covers the entire implanted surface and no indentations are duplicated.Experimental observations by TEM indicate that EUROFER97 steel experienced an increase in population density of cavities as a function of He concentration. The values of distribution density have been quantified, assuming that most cavity sizes were between 1 and 2 nm. The estimate of the calculated distribution density was 9.6 × 10^23^ m^−3^ in zone A, 3.25 × 10^23^ m^−3^ in zone B, and 1.63 × 10^23^ m^−3^ in zone C. These values suggest that the population density is directly proportional to the concentration of He implanted after the annealing heat treatment at 450 °C for 100 h.The EU-ODS EUROFER steel, on the other hand, shows a notable increase in the size of the cavities, which decreases depending on the concentration of He implanted. It should be borne in mind that only cavities larger than 2 nm were taken into consideration; those ones with a smaller diameter were not taken into account due to their low influence on the swelling phenomenon (or volumetric fraction). In zone A, the average size was 4.7 nm with a distribution density of 1.846 × 10^22^ m^−3^ and a swelling of 0.17%. In zone B, the mean diameter was 3.2 nm, its population density was 3.656 × 10^21^ m^−3^, and swelling of 0.12% was calculated, and, finally, in zone C, the average size of the cavities was between 1 and 2 in almost all cases, with some cavities of 4 and even 5 nm. Therefore, its distribution density was calculated analogously to EUROFER97 (9.656 × 10^22^ m^−3^) and its volumetric fraction was almost negligible (0.002%). The effect of the inherent vacancies in the EU-ODS EUROFER steel seems to play a very important role, as it is possible that the annealing temperature chosen favors the mobility of these defects, thus enhancing the creation and growth of He cavities. It has also been observed that their size depends strongly on the concentration of He implanted.It is not possible to conclude which mechanism governs the nucleation and growth of cavities after a process of annealing at 450 °C for 100 h in both materials when only considering the maturation of the Ostwald and migration and coalescence (MC) models. In order to do so, further experiments would be required to study other annealing times and temperatures and different He concentrations in order to obtain a more complete spectrum of cavity evolution or, if not, to establish a model that fits better than those mentioned in the paper.

## Figures and Tables

**Figure 1 micromachines-09-00633-f001:**
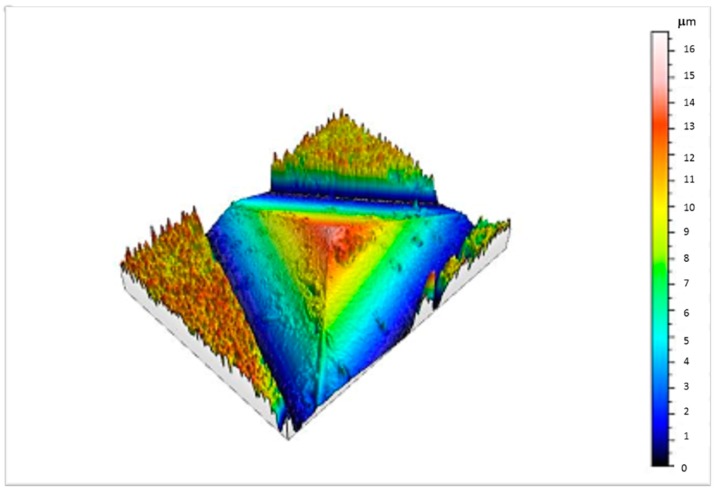
Berkovich indenter tip analyzed by optical profilometer to check the rounding and wear of the very tip.

**Figure 2 micromachines-09-00633-f002:**
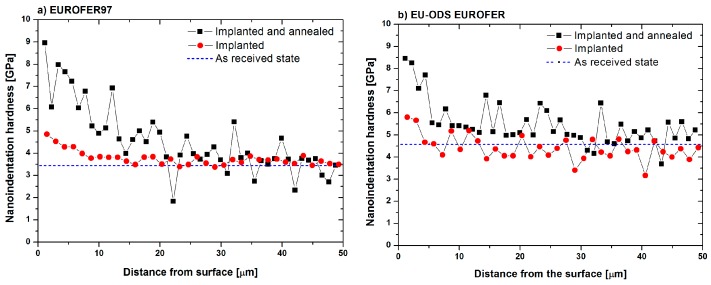
Hardness values vs. distance from the implanted surface of (**a**) EU-ODS EUROFER and (**b**) EUROFER97 irradiated with He from 2 to 15 MeV: Black dotted line represents the values for the annealed state at 450 °C for 100 h. Red dotted are the hardness values for the implanted at RT previous annealing and blue dashed line represents the as received hardness values. All the measurements were performed at 5 mN.

**Figure 3 micromachines-09-00633-f003:**
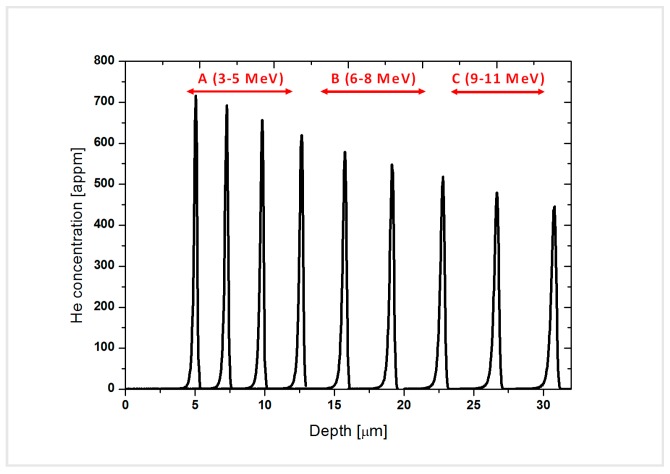
He concentration profiles simulated by MARLOWE indicating the 3 zones in which the lamella was divided for its microstructural characterization by means of TEM.

**Figure 4 micromachines-09-00633-f004:**
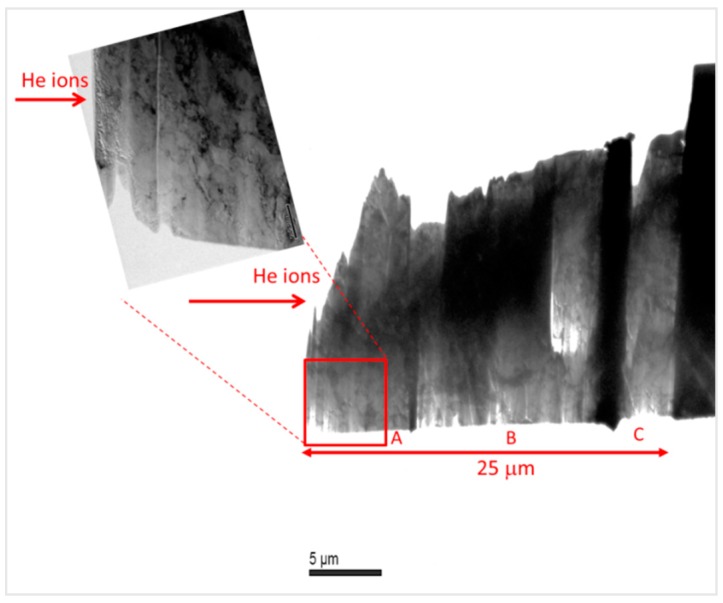
TEM micrograph of the EUROFER97 lamella implanted from 2 to 15 MeV and annealed at 450 °C for 100 h homogeneously thinned along with a detail of the zone A which is the area with the highest He content (716 to 657 appm He).

**Figure 5 micromachines-09-00633-f005:**
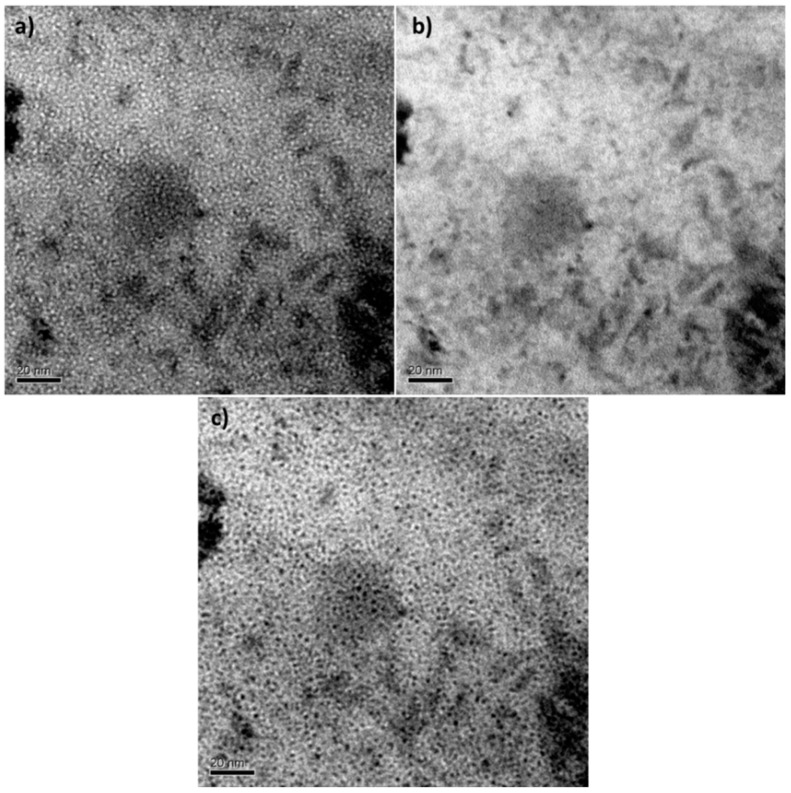
TEM focus sequence of the zone A of EUROFER97 annealed at 450 °C for 100 h and a He content from 716 to 657 appm. The focus variation was about ±1 μm: (**a**) under focused, (**b**) in-focus, and (**c**) over focused.

**Figure 6 micromachines-09-00633-f006:**
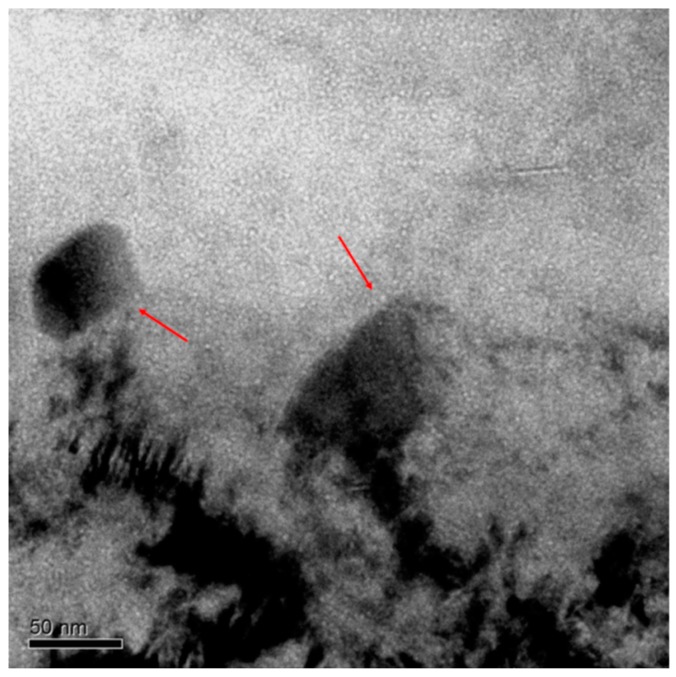
TEM micrograph of EUROFER97 steel annealed at 450 °C for 100 h showing the distribution of bubbles within the matrix in zone B of intermediate He concentration (619 to 548 appm). The arrows indicate bubble nucleation at the M_23_C_6_ matrix interface.

**Figure 7 micromachines-09-00633-f007:**
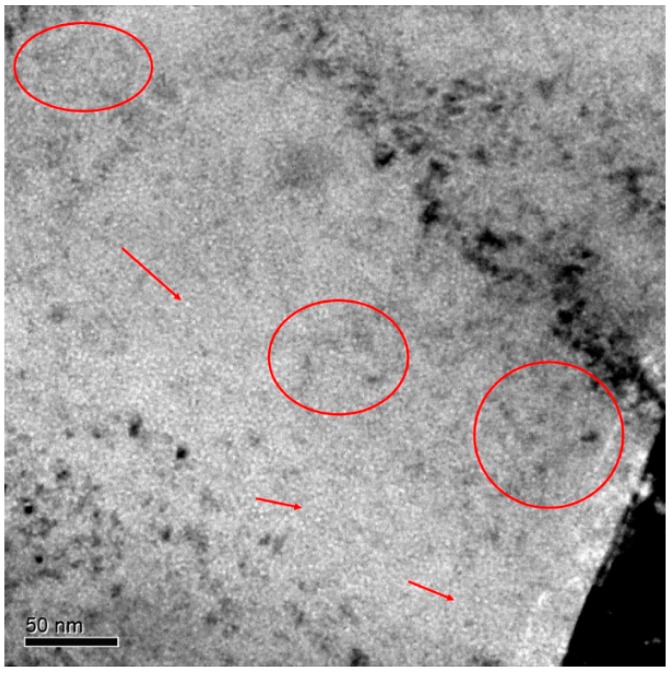
TEM micrograph of EUROFER97 steel annealed at 450 °C for 100 h showing the distribution of bubbles within the matrix in zone C at the lowest He concentration (518 to 445 appm He). Red arrows and ovals highlight some cavities.

**Figure 8 micromachines-09-00633-f008:**
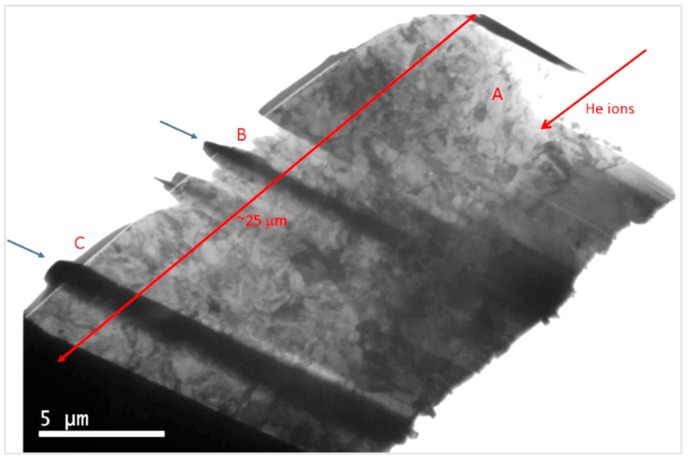
TEM micrograph overview of the lamella obtained from the EU-ODS EUROFER steel annealed at 450 °C for 100 h evenly thinned. The red arrow indicates the implantation surface. A, B, and C indicate areas with different He content and blue arrows show separators.

**Figure 9 micromachines-09-00633-f009:**
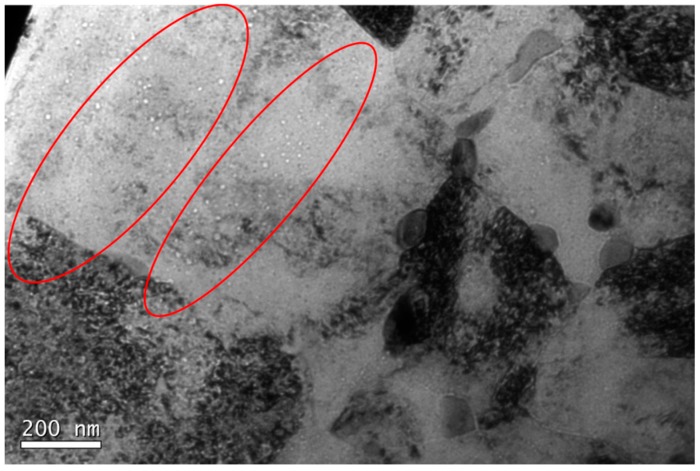
A representative TEM Micrograph of zone A (716 to 657 appm He) of EU-ODS EUROFER steel annealed at 450 °C for 100 h. Highlighted in red the areas of higher cavity density.

**Figure 10 micromachines-09-00633-f010:**
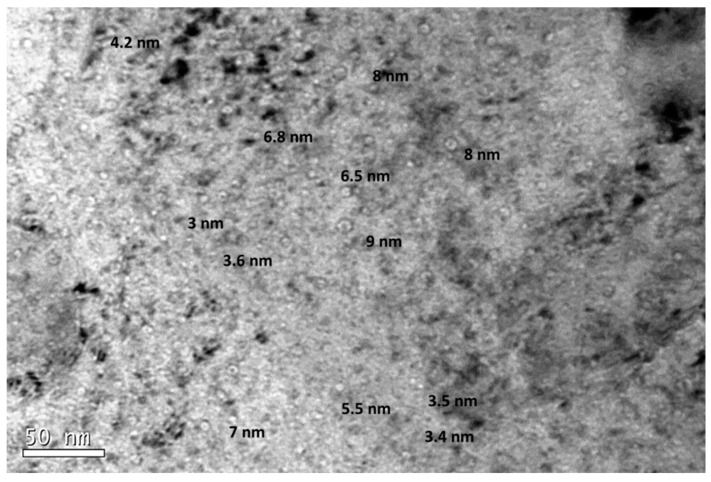
Under-focused TEM micrograph from zone A (716–657 appm He) of EU-ODS EUROFER annealed at 450 °C for 100, including the diameters of some representative cavities.

**Figure 11 micromachines-09-00633-f011:**
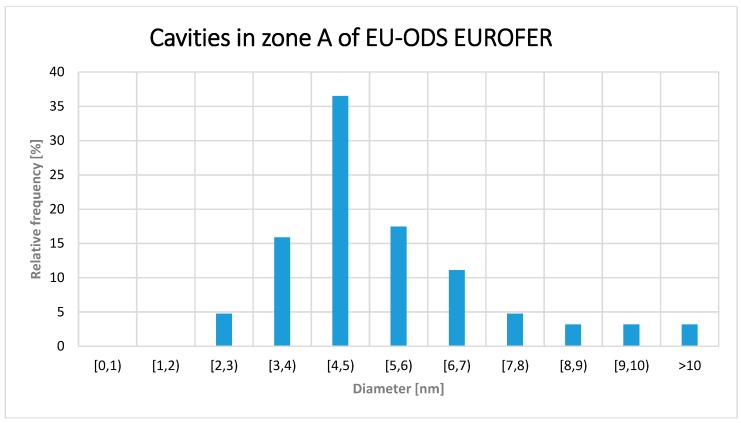
Cavity diameter distribution within the zone A of EU-ODS EUROFER (657–716 appm He) and annealed at 450 °C for 100 h.

**Figure 12 micromachines-09-00633-f012:**
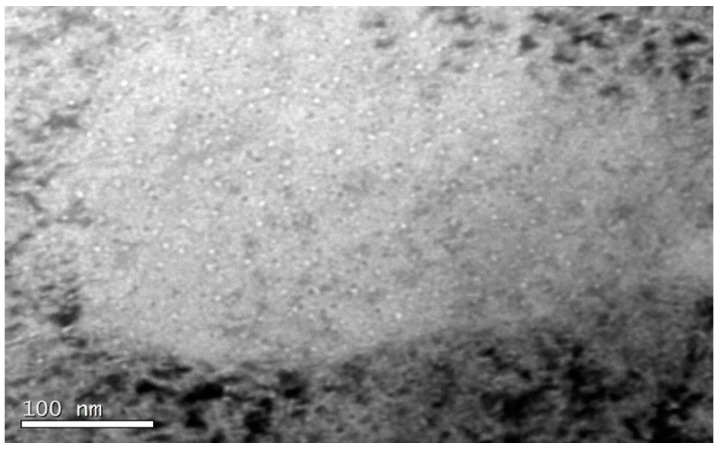
Cavities detected within the zone B of EU-ODS EUROFER (548–619 appm He) annealed at 450 °C for 100 h.

**Figure 13 micromachines-09-00633-f013:**
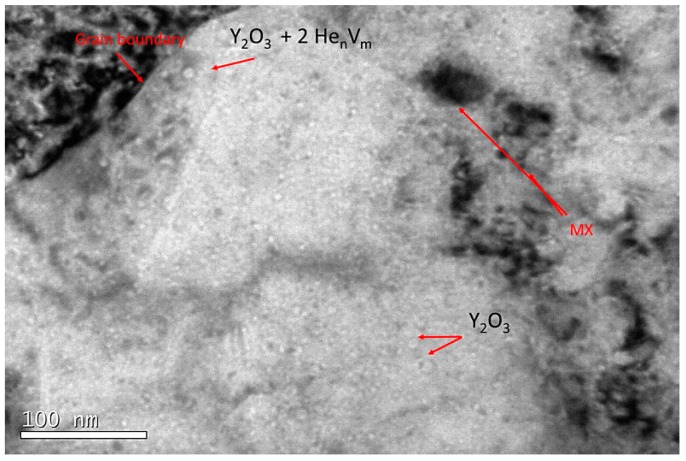
TEM micrograph belonging to zone B (548–619 appm He) of EU-ODS EUROFER steel annealed at 450 °C for 100 h. A grain boundary and two MX type precipitates are highlighted with no clear indication of cavities in their surrounds. In addition, two yttria particles with no cavities along with another one with two cavities attached are also pointed out.

**Figure 14 micromachines-09-00633-f014:**
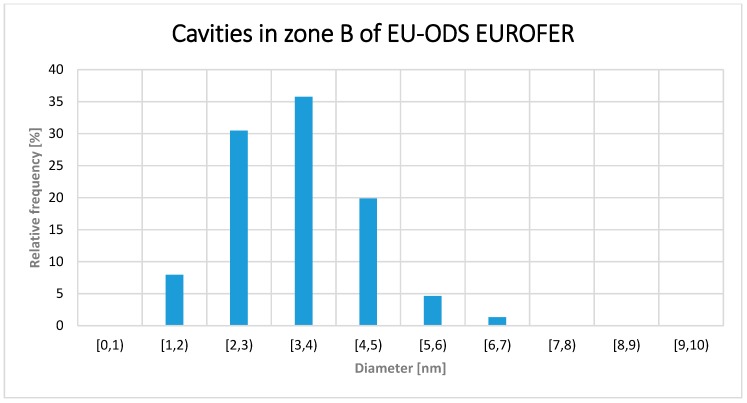
Cavity diameter distribution within the zone B of EU-ODS EUROFER (548 to 619) and annealed at 450 °C for 100 h.

**Figure 15 micromachines-09-00633-f015:**
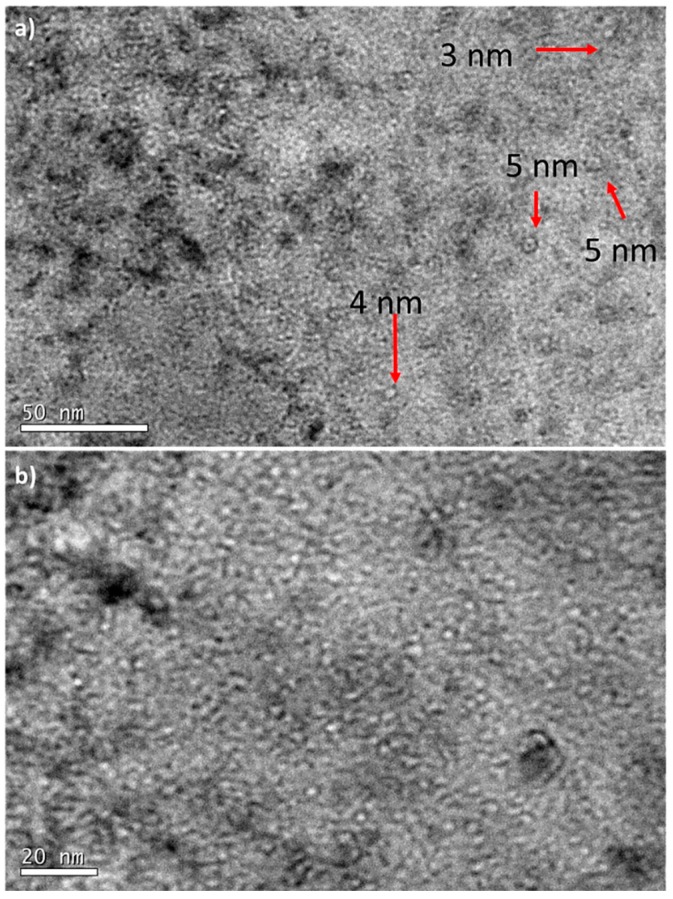
TEM micrograph representative of zone C (518 to 445 appm He) of EU-ODS EUROFER steel annealed at 450 °C for 100 h showing (**a**) very few medium size cavities and (**b**) the overall distribution of small cavities.

**Figure 16 micromachines-09-00633-f016:**
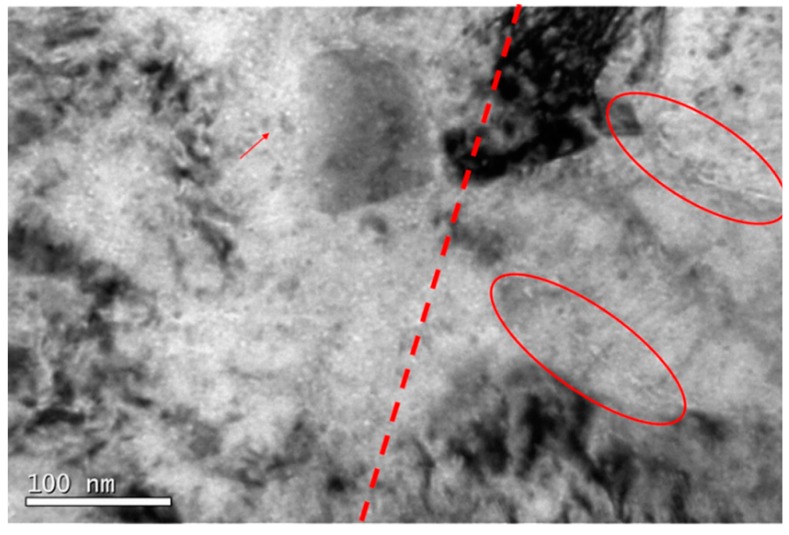
TEM micrograph from zone C of EU-ODS EUROFER steel annealed at 450 °C for 100 h where a possible end of a Bragg peak is indicated as well as different cavity groupings.

**Figure 17 micromachines-09-00633-f017:**
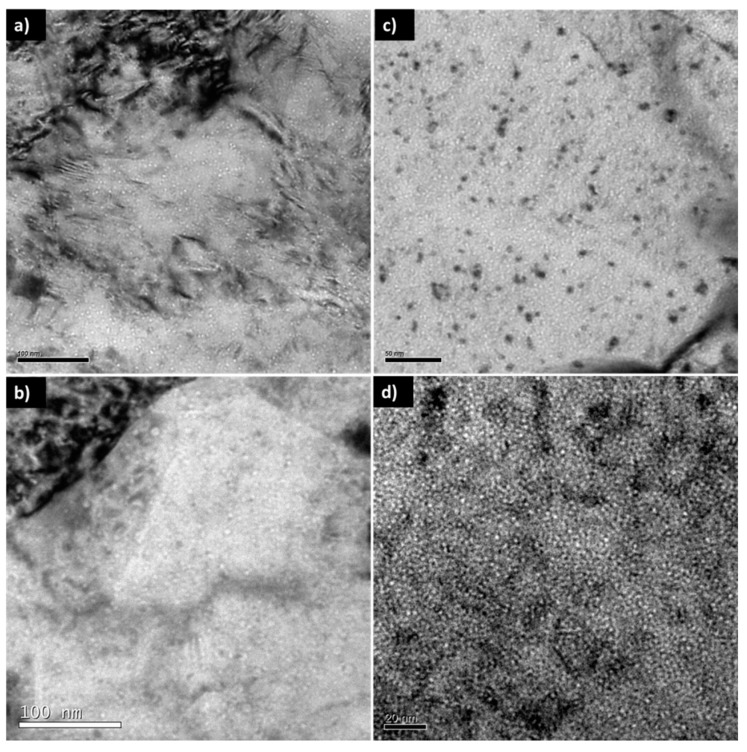
TEM micrographs of the microstructure of EU-ODS EUROFER (**left**) and EUROFER97 (**right**) steel with a He content of ~700 appm He (**a**) and (**c**) implanted at room temperature with stair-like profile from 2 to 15 MeV and (**b**) and (**d**) implanted at room temperature with stair-like profile from 2 to 15 MeV and annealed at 450 °C for 100 h.

**Figure 18 micromachines-09-00633-f018:**
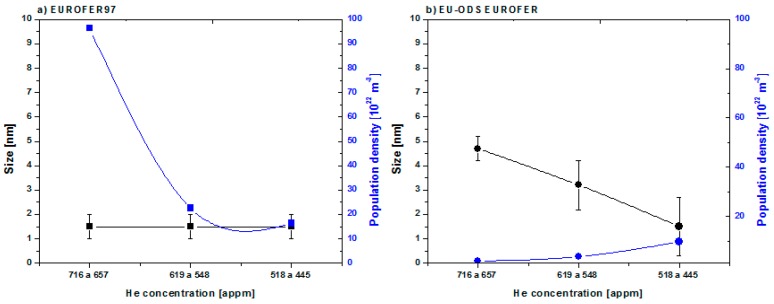
Cavity size and population density measured along the implantation depth profile for (**a**) EUROFER97 and (**b**) EU-ODS EUROFER.

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
