# Peer review of "Nanoindentation and TEM to Study the Cavity Fate after Post-Irradiation Annealing of He Implanted EUROFER97 and EU-ODS EUROFER"

_micromachines, 2018, doi:10.3390/mi9120633_

Round 1

Reviewer 1 Report

In this study the authors use Nanoindentation and TEM to study the effect of He and annealing in EUROFER97 and EU-ODS EUROFER. The scientific results are solid, however, the presentation can be improved. I have the following questions:

1) Is the title correct? "Nanoindentation and TEM to study the cavity fate 2 after post-irradiation annealing of He implanted 3 EUROFER97 and EU-ODS EUROFER with" looks there is a phrase after "with".

2) With annealing implanted, the variation of the nanoindentation becomes large. Why is it? And how does this affect the accuracy of your test?

3) What is the correlation between concentration of He and the annealing? Do they both affect the mechanical properties in a inter-dependent way, or they are two independent effects?

3) There are many figures in this manuscript, but not all of them are in a publishable format. E.g. in Figure 11, the font of the title is different from the axis. And the grey background doesn't contribute to better understanding of the chart.

Author Response

1) Is the title correct? "Nanoindentation and TEM to study the cavity fate 2 after post-irradiation annealing of He implanted 3 EUROFER97 and EU-ODS EUROFER with" looks there is a phrase after "with".

The title is mistaken. Tha last "with" is misplaced.

2) With annealing implanted, the variation of the nanoindentation becomes large. Why is it? And how does this affect the accuracy of your test?

The variation becomes larger since the effect of the irradiation was enhanced due to the temperature effect. In the case of EUROFER97 the temperature promotes the nucleation of bubbles and in the case of EU-ODS EUROFER, the temperature makes the bubbles growth. So, in both cases in larger o lesser extent the annealing temperature enlarge the effect of the implantation at room temperature.

3) What is the correlation between concentration of He and the annealing? Do they both affect the mechanical properties in a inter-dependent way, or they are two independent effects?

There is a sort of synergy between concentration and annealing temperature, since the larger is the He concentration the larger is the hardening due to annealing temperature. However,if the temperature is too much high, the effect of irradiation can be removed totally, along with sever microstructural changes. 

3) There are many figures in this manuscript, but not all of them are in a publishable format. E.g. in Figure 11, the font of the title is different from the axis. And the grey background doesn't contribute to better understanding of the chart.

The figures were changed.

Reviewer 2 Report

Some minor details should be checked.

Lines 18, 93, 165, 172, 190, 210, 271, 360, 362 a symbol before "m" is missing.

Lines 124, 305, 334, 347, 384, 435, 461, 519, 534, 545, 561 please, check, some confusions with the degree of Celcius symbol.

The Figures 7, 12, 13 and 15b, the quality and presentation are not sufficient to understand the results discussed.

Author Response

nes 18, 93, 165, 172, 190, 210, 271, 360, 362 a symbol before "m" is missing.

Micrometer symbol is missing. Replaced!

Lines 124, 305, 334, 347, 384, 435, 461, 519, 534, 545, 561 please, check, some confusions with the degree of Celcius symbol.

Changed.

The Figures 7, 12, 13 and 15b, the quality and presentation are not sufficient to understand the results discussed.

The quality is due to be obtained by FIB. An image enhancement is already used.